# "Yellow" laccase from *Sclerotinia sclerotiorum* is a blue laccase that enhances its substrate affinity by forming a reversible tyrosyl-product adduct

Augustin C. Mot[1,2,3]*, Cristina Coman[1], Niculina Hadade[1], Grigore Damian[4], Radu Silaghi-Dumitrescu[1], Hendrik Heering[3]

**1** Faculty of Chemistry and Chemical Engineering, Babes-Bolyai University, Cluj-Napoca, Romania, **2** Department of Biomolecular Physics, National Institute for Research and Development of Isotopic and Molecular Technologies, Cluj-Napoca, Romania, **3** Leiden Institute of Chemistry, Leiden University, Leiden, The Netherlands, **4** Faculty of Physics, Babes-Bolyai University, Cluj-Napoca, Romania

* augustinmot@chem.ubbcluj.ro

**Data Availability Statement:** All relevant data are within the paper and its Supporting Information files.

## Abstract

Yellow laccases lack the typical blue type 1 Cu absorption band around 600 nm; however, multi-copper oxidases with laccase properties have been reported. We provide the first evidence that the yellow laccase isolated from *Sclerotinia sclerotiorum* is obtained from a blue form by covalent, but nevertheless reversible modification with a phenolic product. After separating the phenolics from the extracellular medium, a typical blue laccase is obtained. With ABTS as model substrate for this blue enzyme, a non-natural purple adduct is formed with a spectrum nearly identical to that of the 1:1 adduct of an ABTS radical and Tyr. This modification significantly increases the stability and substrate affinity of the enzyme, not by acting primarily as bound mediator, but by structural changes that also alters the type 1 Cu site. The HPLC-MS analyses of the ABTS adduct trypsin digests revealed a distinct tyrosine within a unique loop as site involved in the modification of the blue laccase form. Thus, *S. sclerotiorum* yellow laccase seems to be an intrinsically blue multi-copper oxidase that boosts its activity and stability with a radical-forming aromatic substrate. This particular case could, at least in part, explain the enigma of the yellow laccases.

## Introduction

Laccases (benzenediol:oxygen oxidoreductases, EC 1.10.3.2), one of the most widely encountered class of blue copper containing oxidases, have a wide distribution among many fungi, higher plants, insects and bacteria[1]. They catalyze the one-electron oxidation of a broad range of phenolics, aromatic amines and of some inorganic compounds, as well as the four-electron reduction of molecular oxygen to water[2]. Their active sites feature four copper ions grouped into three spectroscopically-distinguishable centers based on UV-vis and EPR (Electron Paramagnetic Resonance) data. The type 1 copper is responsible for the blue characteristic

**Funding:** This work was partly financially-supported by the UEFISCDI Romania, project number PN-III-P1-1.1-PD-2016-0121 to ACM. https://uefiscdi.gov.ro/. The funders had no role in study design, data collection and analysis, decision to publish, or preparation of the manuscript.

**Competing interests:** The authors have declared that no competing interests exist.

color of the protein, with a 600 nm band in the UV-vis spectrum due to an S-Cu LMCT transition (ligand-to-metal charge transfer); it is also characterized by a narrow hyperfine coupling in the EPR spectrum. The type 1 center is the electron acceptor site, where the substrates are oxidized. The type 2 copper gives a normal EPR spectrum but no distinct band in UV-vis spectroscopy, as it only features nitrogen and oxygen ligands. The type 3 center is binuclear; it can be observed in UV-vis as a weak band around 330 nm, but usually gives no EPR spectrum due to the antiferromagnetic coupling between the two metals. Together, type 2 and type 3 form a trinuclear center where the oxygen is reduced to water with electrons received from type 1[3]. Laccases are known for their physiological roles such as lignification, delignification, plant defense, fungal pathogenesis, sporulation and morphogenesis[4], but are also used in various applications such as bioremediation, paper industry, food industry, cosmetics, textiles, biosensors and biocatalysis[5].

In the past three decades, several atypical laccases with unusual spectral properties have been isolated and characterized. They lack the specific 600 nm band and are thus no longer blue—but rather pale yellow, hence the name 'yellow' or 'white' laccases. Despite such differences, the fact that they are multi-copper containing enzymes able to oxidize phenolic compounds, methoxy-substituted phenols, aromatic amines and several others organic compounds but not tyrosine, recommends them as true laccases[6]. Yellow laccases have been highly purified from the common mushroom *Agaricus biosporus*[7], phytopathogenic ascomycete *Gaeumannomyces graminis*[8], *Schizophyllum commune*[9], *Panus tigrinus*[10], *Phlebia radiata*[11], *Phellinus ribis*[12], and *Pleurotus ostreatus* D1[13,14], Trametes sp. F1635[15]. A thermostable and solvent-tolerant laccase from *Ganoderma fornicatum* was obtained as recombinant in *Pichia pastoris* and appeared to be a yellow laccase[16]. Moreover, thalli of lichens *Solorina crocea* and *Peltigera aphthosa* were shown to secret two laccase isozymes—one "blue" and one "yellow"[17]. A series of other non-blue laccases from fungi and prokaryotes have also been purified and characterized[18–22].

Leontievsky and coauthors showed that with the same organism different culture conditions lead to spectral differences of the purified laccases (yellow or blue)[11]. Knowing that the two forms have similar copper content, they proposed that the yellow laccase can form by binding of low-molecular weight phenolic material (i.e. such material is expected to be produced during fungal growth, as part of lignin degradation) to blue laccase. It was previously proposed that bound phenolic substrates are not only responsible for the change in color, but also act as exogenous mediators in the reaction of oxidation of high potential substrates [10,23]. Thus, it was shown that the yellow laccases from *P. tigrinus* and *P. ostreatus* D1 are able to oxidize non-phenolic compounds in the absence of mediators in contrast to their corresponding blue forms[10,13,14]. A yellow laccase was also obtained by site-directed mutagenesis of a blue form preserving all its catalytic functions thus showing that alterations of the T1 pocket could be another explanation for this peculiar property[24]. A colorless laccase, termed as "white laccase", purified from *Pleurotus ostreatus* was reported to contain only one copper atom/molecule with the other two coppers replaced by two zinc and one iron ions.[25] Other 'white' laccases were purified from *Pycnoporous sanguineus*[26], from *Trametes hirsuta*[27], and from *Phlebia radiata* BP-11-2[28]. In the *T. hirsuta* white laccase, one copper atom was replaced by manganese.

The basidiomycete *Steccherinum ochraceum* strain 1833 secrets a laccase with a band at 611 nm, with molar extinction coefficients (7200–7800 $M^{-1} cm^{-1}$) distinctly higher than those of other known "blue" laccases, and a lower spectral shoulder at 330 nm[29]. This suggests modified geometries of the T1 and T3 active sites; and could explain their significantly higher optimal temperatures and thermostability.

The atypical properties of these "non-blue" laccases may allow one to raise the question: are the variations in laccase T1 spectral properties linked to a specific function? We have previously reported on the isolation and characterization of a laccase from *Sclerotinia sclerotiorum*, with two apparently conflicting spectroscopic properties[30]. Although the typical type 1 sulfur to Cu(II) charge-transfer band was absent from the electronic absorption spectrum, thus classifying it as a "yellow" laccase, the EPR spectrum revealed the presence of an oxidized type-1 copper. Here, we provide, for the first time, conclusive evidence that, at least in the case of this *S. sclerotiorum* laccase, the yellow form is obtained from a blue form by covalent modification at the T1 site with metabolites produced by the laccase itself, as previously proposed.[10,11,23] Moreover, this modification significantly improves the catalytic properties of the enzyme. The modification consists in the formation of a reversible adduct between the radical product and a Tyr residue on the enzyme. Because each aromatic model substrate yields an adduct with a distinct color, it would be appropriate to call the enzyme a "rainbow" laccase.

## Materials and methods

### Chemicals

*N*-tris-(hydroxymethyl)-methyl-3-aminopropanesulfonic acid (TAPS buffer), 3-(*N*-morpholino)propanesulfonic acid (MOPS buffer), 2-(*N*-Morpholino)ethanesulfonic acid 4-Morpholineethanesulfonic acid (MES buffer), 3,3',5,5'-tetramethylbenzidine (TMB), guaiacol, 2,3-Dimethoxy-5-methyl-p-benzoquinone (Q0), sodium fluoride, sodium acetate, ammonium persulfate, sodium dithionite, and L-tyrosine, were obtained from Sigma-Aldrich (Steinheim, Germany). 2-Amino-2-hydroxymethyl-propane-1,3-diol (TRIS buffer) and potassium ferricyanide were purchased from Merck (Darmsstadt, Germany). Guanidine hydrochloride was purchased from Alfa-Aesar (Karlsruhe, Germany). Water-free acetic acid and ethanol were purchased from Nordic Invest (Cluj-Napoca, Romania) and sodium chloride from Poch (Gliwice, Poland), while ammonium sulfate was purchased from Lach-ner (Brno, Czech Republic). The compounds and solvents were of HPLC or analytical degree purity. For buffers and washing, ultra-pure water (0.055 µS/cm) obtained via an SG ultra-pure water system (Barsbüttel, Germany) was used. 2,2-Azino-bis-(3-ethylbenzthiazoline-6-sulfonic acid) (ABTS) was purchased from TCI-SU (Tokyo, Japan).

### Laccase purification

The *Sclerotinia sclerotiorum* fungus was cultured in liquid medium as previously described [30]. When the activity reached the maximum level, the extracellular medium was collected and concentrated using an Amicon concentrator (10 kDa-cutoff membrane) under positive pressure followed by dialysis in 20 mM TrisHCl, pH 6.3. The obtained solution was passed through an anion exchange column (HiTrap, GeHealthcare), using a gradient starting with 100% 20 mM TrisHCl pH 7 (buffer A) up to 50% 20 mM TrisHCl, NaCl 1M pH 7 (buffer B). The laccase fractions were eluted at ~27% B. The fractions with maximum laccase activity were concentrated in an Amicon concentrator with a 10 kDa-cutoff membrane and then passed through a hydrophobic interaction column (HiTrap Phenyl, GEHealthcare) after the addition of ammonium sulfate to a final concentration of 1.5 M; The column was washed with buffer C: 20 mM TrisHCl, pH 7, $(NH_4)_2SO_4$ 1.5 M buffer and eluted with buffer A. The laccase was eluted from the column at ~ 90% B. The fractions with maximum laccase activity were pooled together, concentrated as stated above and loaded onto a HiPrep 16/60 Sephacryl S-100 size exclusion column (GE Healthcare) in 20 mM TrisHCl 7.4 pH buffer with 150 mM NaCl. This purification protocol led to a homogenous band on SDS-PAGE corresponding to a molecular weight of ~66 kDa. The purified protein exhibited a typical yellow laccase UV-vis

spectrum. The protein was previously assigned to the predicted oxidoreductase A7EM18 in the *Uniprot* database[30].

In order to obtain the blue form, the previously described chromatographic steps were preceded by a preparative native-PAGE. During this electrophoresis step, the aromatic/phenolic material was seen to migrate before the proteins as a large dark brown band. The dark polyphenolic gel band was cut off and removed, and rest of the gel containing the protein fraction was then chopped and eluted in 20 mM MOPS pH 7 for ~ 1h under stirring, followed by the separation of the gel by centrifugation and filtration (0.22 μm filter). Further elution of the remained gel parts led to a solution with very little laccase activity, thus ensuring that the first elution was almost quantitative. The supernatant was dialyzed overnight in 20 mM MOPS pH 7 and then the previously described chromatographic steps were followed. This protocol led to the blue form of the laccase. Another protocol which leads to blue form is by modification of the culture media. Using the same components as previously described[30] but autoclaved them separately in three distinct jars: salts and trace elements, carbon source and yeast extract. Their mixture after sterilization leads to a much less colored culture media. Then, the purification protocol being followed as previously described.

## Adduct preparation

Adducts of guaiacol, TMB, and ABTS with the blue laccase were formed as follows: 20 μL of 32 μM blue laccase in 25mM MES, pH 6.3 buffer was further diluted with 50 μL of the same buffer, then 10 μL of substrate (guaiacol, TMB, or ABTS) was added to the protein solution to a final concentration of 125 μM. The reaction was allowed to proceed for three minutes, then the mixtures were passed through a Bio-Spin or micro Bio-Spin size exclusion column (Bio-Rad) which was pre-equilibrated with 25mM MES pH 6.3 buffer. Further washing using 50 kDa cut-off Amicons, overnight dialysis or size exclusion columns did not alter the UV-vis spectrum of the protein. When required, higher sample volumes were used but the substrate/enzyme reaction ratio was kept constant. When increasing the reaction ratio, the UV-vis spectrum profile was identical but the purification required additional desalting steps, thus decreasing the yield. For ABTS, three additional preparations of adducts were performed: two by replacing the reduced ABTS with preformed $ABTS^{+\bullet}$ and $ABTS^{2+}$ respectively, and one by extra addition to the reaction mixture of 90 mM NaF to inhibit $O_2$ reduction at the trinuclear site.

## UV-vis and fluorescence measurements

UV-vis spectra were measured on a Varian Carry 50 spectrophotometer in either a 100-μL ultra-micro quartz cuvette or in a 1 mL quartz cuvette, depending on amount of protein sample available. The protein samples were in 25 mM MES pH 6.3 buffer. GuHCl (guanidinium hydrochloride) titration of the ABTS-laccase adduct was done in the same cuvette and buffer, by the addition of increasing amounts of GuHCl from a 6M stock up to a GuHCl concentration of 5.2 M. The measurements were done in duplicates or triplicates. For determining the enzymatic parameters, 2,3-dimethoxy-5-methyl-p-benzoquinol ($Q_0H_2$), ABTS and 3,3',5,5'-tetramethylbenzidine and guaiacol were used as substrates in 1 mL 50 mM acetate pH 4 buffer and with 6.3 nM laccase adduct. Reduced $Q_0$ ($Q_0H_2$) was prepared by electrochemical reduction of an anaerobic 25 mM stock solution of the bezoquinone. A 0.9 ml cup-shaped stainless-steel working electrode was used, stirred with a small magnetic stirrer bar, and closed by a lid incorporating a vicor-glass salt bridge to a saturated KCl solution in contact with both an Ag/AgCl reference electrode and a Pt counter electrode. The WE potential was held at -0.8 V for 1.1 hours by a microAutolab II potentiostat. The UV-vis spectrum confirmed the quantitative

reduction of the benzoquinone. The substrate oxidation was monitored as follows: formation of $ABTS^{+\bullet}$ at 420 nm using 36000 $M^{-1}cm^{-1}$ as extinction coefficient, oxidation of guaiacol monitored at 465 nm using 12000 $M^{-1}cm^{-1}$ as extinction coefficient, formation of oxidized $Q_0$ at 410 nm using 770 $cm^{-1}M^{-1}$ as extinction coefficient, oxidation of 3,3',5,5'-tetramethylbenzidine at 290 nm using 21000 $M^{-1}cm^{-1}$ as extinction coefficient[46]. The fluorescence spectra were measured using a Perkin Elmer LS55 fluorescence spectrometer. For temperature dependence of the activity, the UV-vis spectrophotometer was coupled to a water-based thermostat (Julabo Labortechnik GmbH, Seelbach, Germany) and the cuvette was placed in a Cary 50 multicell holder adaptor. For the thermal stability of the laccase, 200 μL of 20–30 μM enzyme in 20 mM, pH 6.8 TrisHCl buffer was incubated at the desired temperature and a proper aliquot was removed at a time interval for activity measurement.

## EPR measurements

For EPR spectra, a Bruker EMX Micro spectrometer with a liquid nitrogen cooling system was employed. Instrument conditions were: microwave frequency 9.43 GHz, microwave power 15.89 mW, modulation frequency 100 kHz, modulation amplitude 3 G, sweep rate 22.6 G/s; time constant 81.92 ms, average of three sweeps for each spectrum, temperature 100 K.

## HPLC-MS measurements

A volume of 5.5 mL of 6 μM blue, yellow or ABTS-laccase adduct in sodium phosphate buffer 50 mM, pH 7.5 was incubated with PNGase F (BioLabs) at a final concentration of 1500 U/mL at 37°C for 48 h. This time was required to unsure sugar units removal since the substrate for PNGase was a non-denatured protein. The PNGase F (36 kDa) was separated from laccase (65 kDa) using a 50 kDa Amicon filter by first concentration followed by washing of the reaction mixture with four 5 mL of 10 mM ammonium bicarbonate pH 8. The final volume collected was about 200 μL, which was treated with 10 μL trypsin (0.1 μg/mL) and incubated for 6 h at 37°C. The resulting peptides were further analyzed by HPLC-MS.

HPLC-MS analysis was performed on an Accela$^{TM}$ High-Speed-LC system, equipped with UV-VIS detection, and coupled to a high resolution LTQ ORBITRAP XL mass spectrometer (ThermoScientific). Liquid chromatography separations were achieved on a C18 VYDAC (4.6 mmx150 mm, 5 μm) column, eluting with a linear gradient of 10%-50% acetonitrile/water (containing 0.1%TFA) for 40 min; total run time 55 min; flow rate 1 mL/min; injection volume 20 μL or a Hypersil Gold, ThermoScientific (2.1 mm x 45 mm, 1.9 μm) column, eluting with the following gradient: 3%-10% acetonitrile/water (0.1%TFA) for 4 min., 10%-20% acetonitrile/water (0.1%TFA) for 10 min and 20%-45% acetonitrile/water (0.1%TFA) for 5 min.; total run time 25 min; flow rate 0.25 ml/min; injection volume 2 μL. The retention time ($R_t$) is given in minutes with the gradient in percentage of acetonitrile. Absorbance was monitored at 230 nm, 280 nm and 555 nm. Autosampler and column temperature were set to 5°C and 25°C, respectively. High Resolution Mass Spectra (HRMS) were obtained using positive-ion mode electrospray (ES+) ionization technique. The instrument was externally calibrated using the manufacture's ES(+) calibration mix. The following ionization conditions were used: spray voltage (5 kV), sheath and auxiliary gas flow (40 and 7 arbitrary units, respectively), capillary temperature (275°C), capillary voltage (45 V), tube lens voltage (+245 V). The microscans were set to three.

## Tyrosine targeting experiments

A volume of 100 μL 10 μM laccase in 500 mM sodium citrate pH 5, containing 150 mM sodium nitrite and 150 mM hydrogen peroxide (both added in three successive steps at two

hours' interval) which produce peroxynitrite, was incubated 10 hours in order to nitrate the tyrosine residues of the protein. After salts removal by micro Bio-Spin size exclusion columns (Bio-Rad), the laccase assessed for its activity and was treated with ABTS in order to test its ability to form adducts.

The ABTS-tyrosine adduct was obtained by mixing 500 μM tyrosine and 500 μM ABTS$^{+\bullet}$ in presence of catalytic laccase (7 nM) was incubated overnight in 50 mM ammonium acetate pH 5 and then analyzed by HPLC-MS system using the same method as previously described. A solution of 500 μM (tyrosine concentration) stoichiometric mixture of the aminoacids which from the identified modified peptide (TANNTNP**Y**TNPPNTGVIR) with and without tyrosine was incubated with 500 μM ABTS$^{+\bullet}$ and the reaction was monitored overnight in 20 mM sodium citrate pH 4.5 and in the presence of a catalytic amount of laccase (7 nM). Bovine serum albumin (BSA) was also used as control test, in the same manner.

## Results and discussions

### Blue laccase isolation

Most of the previous reports on a "yellow" laccase proposed that the type 1 spectrum might be quenched by a bound aromatic compound,[10–14] and that this exogenous aromatic compound now acts as a redox mediator, allowing the enzyme to oxidize even high redox potential compounds.[10,13,14,23] To our knowledge, definitive proof for this hypothesis has not been reported. An alternative explanation, according to which distortions of the coordination geometry of the T1 site could lead to bleaching of the 600-nm band, could not be confirmed by computations of the respective UV-vis spectra[31]. We now report that using a preparative native PAGE step before the chromatographic procedures (anion exchange, hydrophobic and size exclusion), one can indeed separate the enzyme from the phenolics or other aromatic compounds, thus yielding a typically blue laccase. If no native PAGE is performed, the enzyme purified from the same crude extract is yellow (Fig 1).

The UV-vis spectrum of the blue laccase presents a slightly higher A$_{280/250}$ ratio (2.1 vs. 1.7) and it can be observed that there is an extra contribution in the yellow laccase UV-vis spectrum bellow 440 nm, most probably due to an aromatic molecule bond to the protein (Fig 1). In addition, the ratio of the absorbances at 280 nm to 600 nm decreased from 125 (yellow form) to 34 (blue form), indicating a dramatic modification of the type 1 copper center, for the as purified forms. This ratio is still higher than most of the multicopper oxidases (cca. 22) and has to be taken with caution, since both, the oxidation state and integrity of the type 1 copper center of the as purified forms could be fully (in yellow case) or partly (blue form) affected due to a putative bound mediator. However, one may ensure that the A$_{280/250}$ is high enough since the presence of high amounts of simple phenolics might obscure the 600 nm band. In our case this ratio was higher (1.66) than in the case of other yellow laccases, even close to other blue ones, thus excluding the existence of such contaminants[32]. Indeed, other purified yellow laccases such as *Pa. tigrinus* laccase and *Ph. radiata* laccase possess lower A$_{280/250}$ ratio (0.95 vs. 0.97 and 0.83 vs. 1.84 respectively) compared to the blue form from the same organism[11]. For these laccases, the same organism leads to yellow or blue form depending on culturing. Solid phase culturing based on wheat straw leads to yellow laccase, while submerged liquid culturing leads to the blue form for both *Pa. tigrinus* and *Ph. radiata*[11].

Since the blue laccase was obtained from the yellow form, our findings appear to confirm the hypothesis that the yellow form is the result of a reaction of the blue form with low molecular weight aromatic compounds, as proposed by Leontievsky and co-workers. Moreover, we report here that, as in the case of Leontievsky[11], culturing *S. sclerotiorum* in low colored (less phenolic/ aromatic material) leads to a blue form of the laccase, with molecular mass and chromatographic

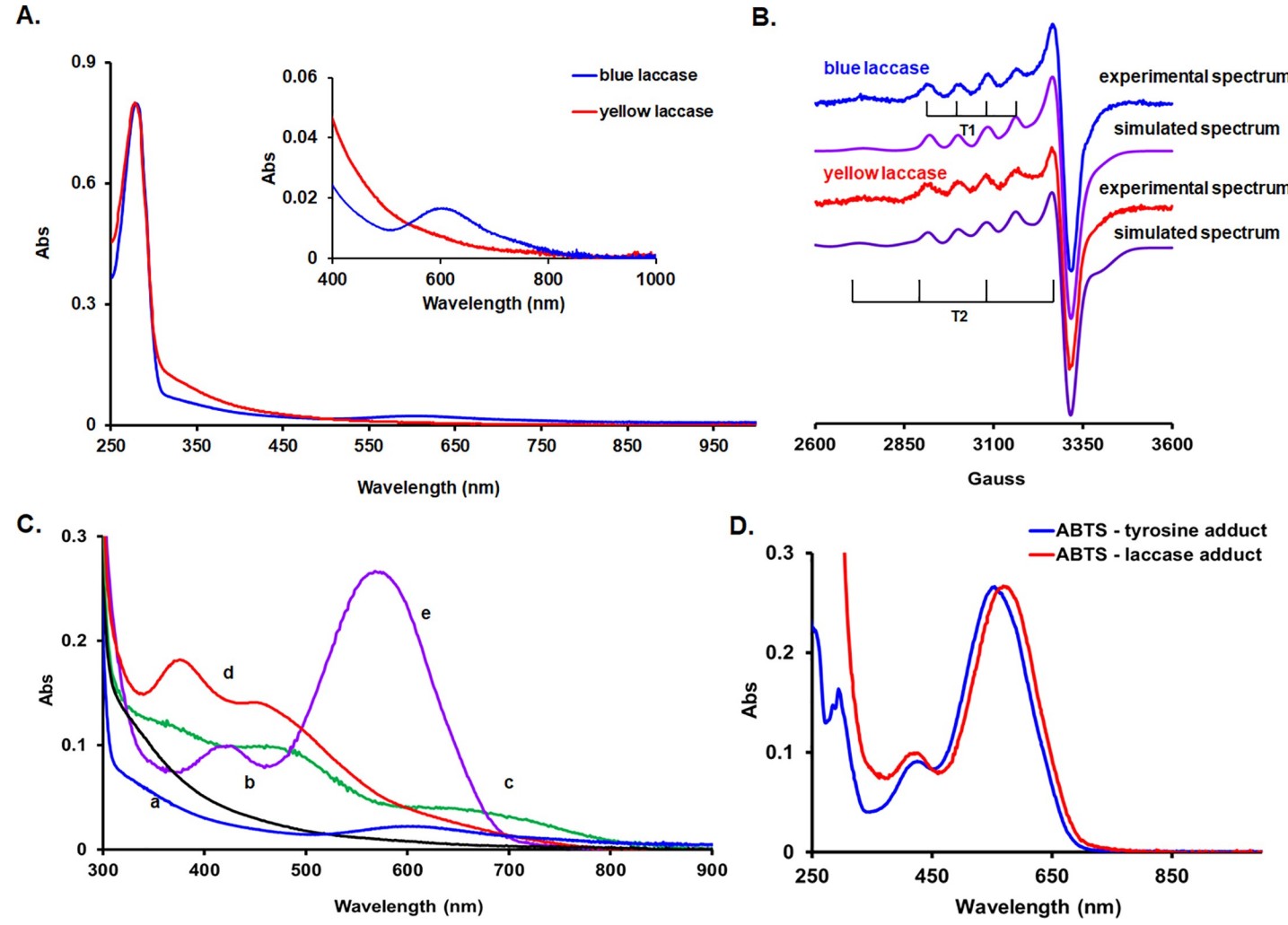

**Fig 1. UV-vis molecular absorption and EPR spectra of the blue, yellow and artificial adducts of the *S. sclerotiorum* laccase. A.** UV-vis spectra of blue and yellow forms of the laccase of *S. sclerotiorum* in 25 mM MES buffer pH 6.3. The protein concentration is 7.5 μM in both cases. **B.** EPR spectra of blue and yellow laccase forms revealing both the type 1 copper center and the type 2 copper center. Conditions: 25 mM MES pH 6.3, 60 μM laccase. **C.** UV-vis spectra of the blue (a) and yellow (b) laccases, and of the red guaiacol (c), orange TMB (d), and purple ABTS (e) laccase adducts. The spectra are normalized at 280 nm. The approximate concentration of the protein (possible $\varepsilon_{280nm}$ variation) is 8 μM. **D.** ABTS-tyrosine (blue) and ABTS-laccase (red) UV-vis spectra at pH 6.3 (25 mM MES).

behavior identical to the yellow form. As shown in Fig 1, the blue form of the *S. sclerotiorum* laccase has a typical EPR spectrum, with the well-known characteristics of both the type 2 and the type 1 copper centers. The spectrum could be simulated by $g_\parallel$ = 2.216, $g_\perp$ = 2.044, $A_\parallel$ = 82 x $10^{-4}$ cm$^{-1}$ for the type 1 copper, and $g_\parallel$ = 2.240, $g_\perp$ = 2.065, A = 189 x $10^{-4}$ cm$^{-1}$ for the type 2 copper which is similar to a reference $[Cu(NH_3)_4]^{2+}$ complex ($g_\parallel$ = 2.240, $g_\perp$ = 2.058, $A_\parallel$ = 181 x $10^{-4}$ cm$^{-1}$). These spectral parameters are well comparable with those obtained for the yellow form (Fig 1)[30] ($g_\parallel$ = 2.217, $g_\perp$ = 2.050, $A_\parallel$ = 83 x $10^{-4}$ cm$^{-1}$ for the type 1 copper, and $g_\parallel$ = 2.240, $g_\perp$ = 2.065, A = 178 x $10^{-4}$ cm$^{-1}$ for the type 2 copper); A value of 189 x $10^{-4}$ cm$^{-1}$ for $A_\parallel$ is in good agreement with other blue laccases data ($A_\parallel$ ~180–195 x $10^{-4}$ cm$^{-1}$)[33].

## Substrate-specific laccase adducts colors

The key observation is that the blue *S. sclerotiorum* laccase is rapidly converted to a deep purple form upon turning over ABTS. In fact, turnover of radical-forming substrates results in a

distinct and substrate-specific color change (purple with ABTS, red with guaiacol, orange with TMB) and significant alteration of the UV-Vis spectrum, as can be observed in Fig 1. We find that this requires exposure of the enzyme to the radical form of the substrate, either upon turnover of the substrate or by the addition of a chemically pre-formed radical to the fluoride-inhibited enzyme. Ferrocenes (non-radical laccase substrates) or 2-electron oxidized ABTS do not modify the enzyme (S1 Fig). With the quinol-type substrate $Q_0H_2$ even after prolonged turnover (tens of minutes), only a very small fraction of modified enzyme can be detected in the difference spectrum, which is in line with the thermodynamic instability of the formed radical (quinones are thermodynamically almost purely two-electron mediators). The EPR spectra of these radical-modified forms of the laccase are very similar to each other, and similar to the blue and yellow forms discussed in the previous paragraphs; the small hyperfine coupling constant, specific to the type 1 copper, is comparable in all cases ($A_\parallel$ is $79 \times 10^{-4}$ cm$^{-1}$ and $g_\parallel$ is 2.216 for ABTS-laccase compared with $A_\parallel = 82 \times 10^{-4}$ cm$^{-1}$ and $g_\parallel$ 2.216, for blue form and $A_\parallel = 83 \times 10^{-4}$ cm$^{-1}$ $g_\parallel = 2.217$ for yellow laccase).

The radical-modified laccase was found not to be altered during the chromatographic separation (anion exchange, size exclusion), dialysis or Amicon concentration (50 kDa cut-off), which suggests that the chromophore is bound to the protein. The adduct binding is very tight: it does not decrease significantly upon buffer exchange, urea or guanidine hydrochloride (7 M) treatment followed by dialysis or desalting. This suggests a covalent attachment, possibly via radical chemistry. However, on a timescale of hours, slow exchange of the TMB adduct with ABTS occurs (but not the reverse), and as mentioned above, the naturally isolated yellow form can be rendered blue by a purification that involves native PAGE to separate the protein from polyphenols prior to any other chromatography step. Furthermore, the deep purple ABTS-laccase adduct is unstable during native PAGE: its color could be observed for only ~10 minutes, after which it disappeared gradually to completeness, presumably decomposed via redox cycling. Native electrophoresis is known for its reduction/oxidation reactions which can induce artifacts in redox active proteins[34]. Likewise, the adduct-containing protein was found to be reduced to a colorless state by dithionite and partly reoxidized when air exposed; subsequent desalting of the reduced form and rapid reoxidation by air allowed only ~50% of the adduct to be regenerated. All these data suggest that the adducts (the artificial ones and the natural yellow one) are very stable but reversible during redox cycling or upon turnover with a 'stronger' radical and suggests that the yellow form is a natural adduct (*vide infra*, section 3.5).

## ABTS$^\bullet$ catalytically binds to a Tyr residue

The UV-vis spectrum of the ABTS-laccase adduct is very similar to that of the ABTS-tyrosine adduct, as can be observed in Fig 1. This suggests that a tyrosine is mostly expected to be the residue that is involved in adduct formation in our laccase. Differences in the spectra ($\lambda_{max} = 575$ nm for laccase adduct and $\lambda_{max} = 555$ nm for free Tyr adduct) might be due to the influence of the protein environment. It is known that *in situ* generated peroxynitrite nitrates exclusively the tyrosine residues[35]. Such modified laccase forms only about 20% adduct compared to the unmodified laccase tested in the same conditions, as deduced from the UV-vis measurements (S2 Fig). This result additionally suggest that the residue involved in the adduct formation is expected to be tyrosine indeed.

Åkerstrom and co-workers[36] suggest that such tyrosine-ABTS adducts are formed on the ortho-position by radical-radical combination. The purple ABTS adduct is formed rapidly with the *S. sclerotiorum* laccase (maximum three minutes), but only slowly (hours) with free monomeric tyrosine (in the latter case, a catalytic amount of laccase to generate and maintain the ABTS radical was part of the reaction mixture) or even slower with BSA (bovine serum

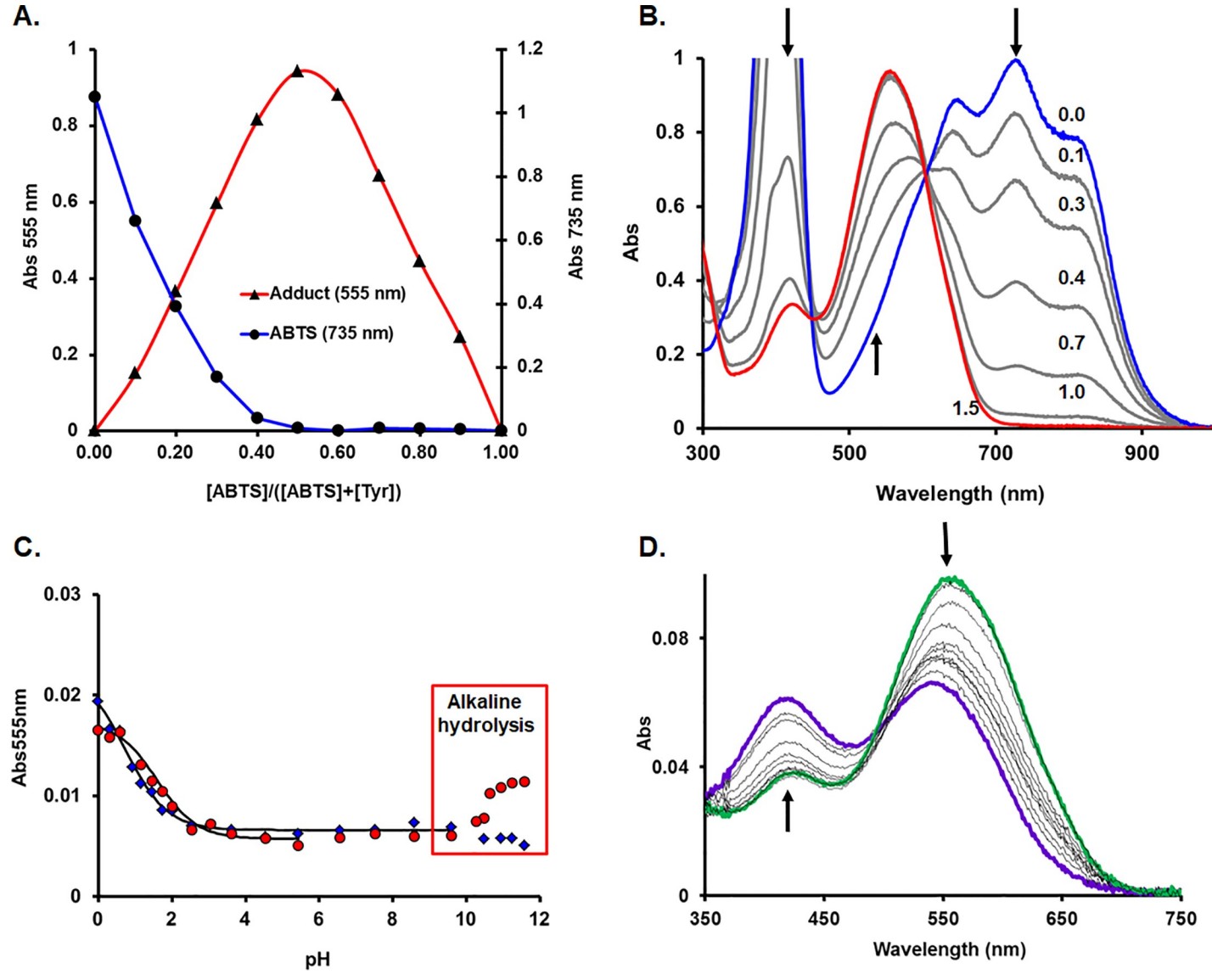

**Fig 2. Free tyrosine—ABTS• interaction and pH titration. A.** Tyrosine and ABTS• form an adduct in a 1:1 ratio, as deduced from a Job's plot. **B.** The solutions' spectra are measured after overnight reaction of ABTS• with tyrosine in different molar reaction ratios ([Tyr]/[ABTS]) as labeled, in presence of 15 nM laccase (to ensure radical form of the ABTS), 50 mM citrate buffer, pH 4. **C.** pH titration of ABTS-tyrosine (circles) and ABTS-laccase (rhombuses) adducts. The pKa values are calculated by fitting the experimental data with a sigmoidal model (see text). **D.** UV-vis changes of ABTS-laccase adduct while pH decreases.

albumin) which leads to an adduct of only 20% intensity at 555 nm compared with the laccase one despite their similar mass and tyrosine composition. Interestingly, this suggests that formation of the adduct is a specific catalytic function of the enzyme. Incubation of free tyrosine with laccase-produced ABTS radical leads to decreases of the ABTS UV-vis features 735 nm and 415 nm and an increase of the 555 nm band specific to ABTS-tyrosine adduct[36]. From tests run at various ABTS/tyrosine ratios, it can be inferred that the adduct is formed in a 1:1 ratio, see Fig 2.

In pH titrations, the ABTS-laccase and ABTS-tyrosine adducts are similar but not identical (Fig 2). Both adducts present a low pKa value (1.51 ± 0.11 for ABTS-tyrosine and 0.73 ± 0.18 for ABTS-laccase), probably due to the sulfonate group. However, at pH higher than 10, the

ABTS-tyrosine adduct suffers irreversible transformation most probably due to hydrolysis, while the ABTS-laccase adduct is stable up to pH 12 (Fig 2). Together with the slight redshift of the spectrum, this suggests that the ABTS-laccase adduct is not fully water-exposed but at least partially shielded by the peptide environment.

## Identification of a key Tyr residue and mechanism of formation

The *S. sclerotiorum* laccase purified by us was successfully assigned to be the putative uncharacterized A7EM18 form *Uniprot* database by MS analysis[30]. Trials to identify a putative adduct site (with a natural substrate) in mass spectrometry analysis according to the protocol used in our previous work[30] failed with all adduct forms. The reductive conditions necessary for disulfide bond breakage (with either DTT or TCEP) followed by iodoacetamide treatment were not possible to follow because this induces adduct breakdown in a similar way to that observed in our dithionite, ammonium persulfate, and electrophoresis experiments (*vide supra*). Thus, we changed the protocol and worked on the peptides obtained by trypsin digestion directly on the deglycosilated ABTS-laccase without disulfide bonds breakage. However, even with these mild conditions protocol, about quarter of adduct was lost (based on partial discoloration). All the other artificial adducts and the yellow laccase seemed to be unstable enough to prevent identification of the site location of the putative adduct, based on our hypothesis. HPLC-MS experiments of such tryptic digestion of pure blue laccase allowed us to identify more than 60% of the predicted peptides (could be much higher if the N-terminal sequenced is removed as posttranslational modification as is happening in all ascomycete laccases). The unidentified peptides could be explained as interference of disulfide bonds or unknown posttranslational modification. Identical LC-MS analyses of tryptic digestion of ABTS-laccase adduct led to identification of all peptide peaks identified in case of pure blue laccase except of only the double charged specie at *m/z* 973.47. This peak was assigned to the following peptide [58]TANNTNP*Y*TNPPNTGVIR[75] (S3 Fig). Moreover, a distinct additional peak was observed in the chromatogram of ABTS-laccase peptides separation which shows in MS spectrum two peaks corresponding to two double charged species at *m/z* = 1228.46 and *m/z* = 1107.97 and which is by far the dominant peak from the entire chromatogram monitored at λ = 555 nm (Fig 3). A few small extra peaks could be observed in the 555 nm monitored chromatogram most probably due to nonspecific ABTS-modified peptides.

The fact that to this peak there are two corresponding masses led to the idea that there are two distinct peptides which overlap. This was proved by using a higher resolution column (Hypersil Gold, ThermoScientific), which better separates the two peptides in two different peaks with the two mentioned masses (Fig 3) in a 10:1 ratio (determined from chromatogram peak area). The largest peak (m/z 1228.46, z = 2) corresponds to the identified peptide with the whole ABTS molecule attached to it (adduct 1) while the other one (m/z 1107.97, z = 2) belongs to the same peptide with just half of the ABTS molecule bond to it (adduct 2) (Fig 3), obtained most probably by cleavage of the first one. These interesting results show that in our case the whole ABTS-laccase adduct is very stable and the ABTS molecule is prevented from breakage due a specific peptide environment in contrast to Åkerstrom and co-workers[36] case, where only the second adduct of ABTS was detected in their glycoprotein. We generated Tyr-ABTS adduct models at several pH's (3–7) but all were detected by HPLC-MS analysis to have only half of the ABTS molecule as in case of our second adduct and the by-product (S4 Fig). All these results are in good agreement with Åkerstrom and co-workers'[36] findings and consistent with the comparative hydrolysis experiments in section *3.3.*, proving that the stabilization of the ABTS-laccase adduct (main adduct, adduct 1) is specific for this site and the modified tyrosine is protected from breakage by proper peptide environment. Furthermore, the

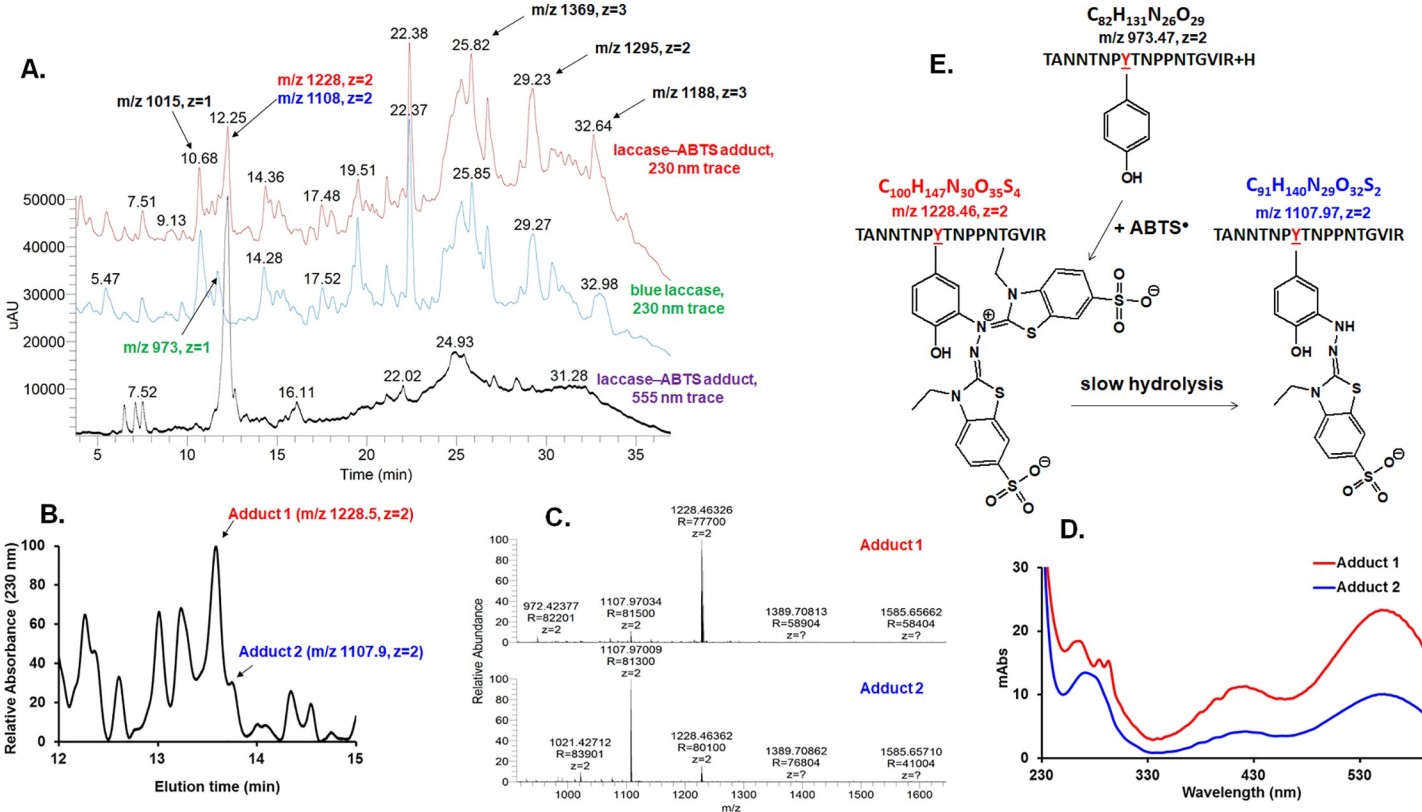

**Fig 3. MS analyses of the ABTS-laccase adduct. A.** HPLC chromatograms of the tryptic digests of blue laccase and ABTS-laccase as monitored at 230 nm and 555 nm. The most important peptides peaks are indicated by arrows and their corresponding m/z and charge. The 11.5 min peak corresponding to the peptide that was present in the blue form but modified in the laccase-ABTS adduct has m/z 973.47, z = 2. The 12.25 min peak corresponds to the modified peptide (two merged adducts, m/z 1228.46 and 1107.97, z = 2 is also indicated. **B.** Separation of the two adducts of the same modified peptide using a high-resolution column. **C.** The MS of the two adducts. **D.** The UV-vis spectrum of the two adducts. **E.** Chemical formulas and proposed structures of the two adducts formation and tentative mechanism of formation.

UV-vis spectrum of the adduct 2, which contains half of the ABTS molecule, has very similar if not identical spectral features in the visible domain with those of Tyr-ABTS molecule—while the UV-vis spectrum of adduct 1, which contains the whole ABTS molecule, has two additional peaks close to 290 nm which can be found in the UV-vis spectrum of the byproduct obtained in the reaction between Tyr and ABTS (Fig 3 and S5 Fig). This brings additional important information regarding the structure of the adducts, thus supporting the MS-based structures.

## Yellow laccase is most probably a natural phenolic-laccase adduct

When the ABTS-laccase adduct was reduced with dithionite or over-oxidized with ammonium persulfate and afterwards desalted, a partial loss of the ABTS-Tyr spectral features was observed. This suggests that strong oxidants as well as strong reducing agents are able to cleave the enzyme-chromophore adduct via redox or radical mechanism; this may explain the ability of ABTS radicals to slowly (hours) replace the TMB adduct, and the removal of the adduct during native PAGE (*vide supra*). On the other hand, guanidine hydrochloride (GuHCl) addition to the laccase-ABTS adduct even close to 7 M—followed by desalting or dialysis—does not fully remove the adduct, even when the spectral changes of the 280 and 550 band suggest full denaturation. After refolding the apo-protein by desalting, ~50% of the adduct is preserved but without any activity. The difference spectrum (before minus after GuHCl treatment,

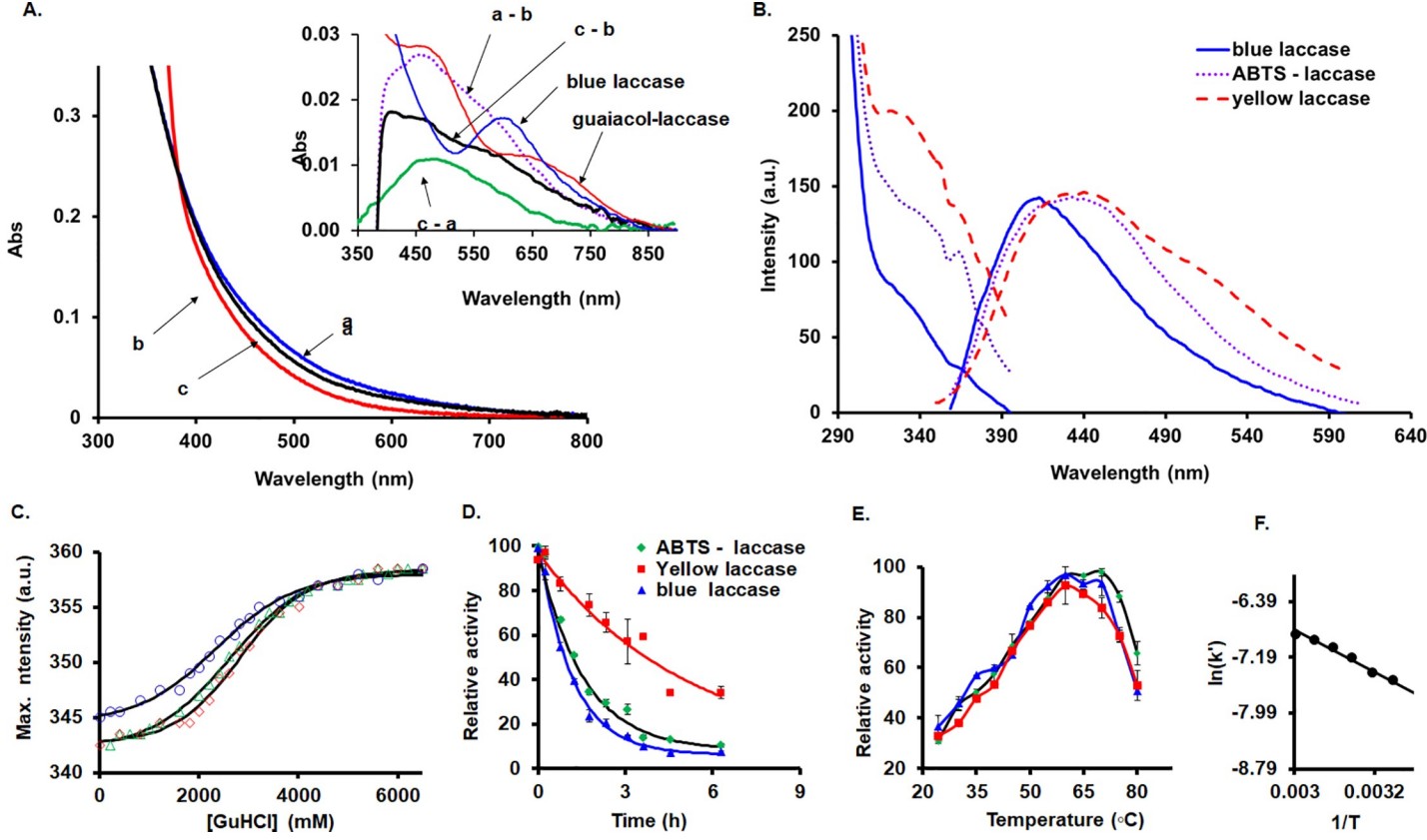

**Fig 4. Molecular absorption and fluorescence spectroscopic investigation of the yellow and artificial adduct-laccase forms. A.** Reduction of pure yellow laccase (a) with dithionite in excess (b) and re-oxidation by exposition to air (c) monitored by UV-vis spectroscopy. Inset: Blue form (solid black), guaiacol-laccase form (solid gray) and difference spectra for yellow laccase (dotted): a–b (isolated oxidized–reduced), c–b (re-oxidized–reduced), and a–c (isolated oxidized–re-oxidized). **B.** Fluorescence spectra (emission while exciting at 330 nm and excitation while followed at 420 nm) of blue, yellow and ABTS-laccase. Protein solutions are in 20 mM TrisHCl pH 6.8. **C.** Titration of yellow, blue and ABTS-laccase with guanidine hydrochloride monitored by the shift of the maximum band of the emission spectrum while the sample was excited at 280 nm, using fluorescence spectroscopy. The experimental data was fitted with sigmoidal curves, yielding inflection points: yellow laccase (circles) 2360 ± 67; ABTS-laccase (triangles) 2616 ± 36; blue laccase (rhombuses) 2835 ± 57. **D.** Exponential decays of activity of yellow, blue and ABTS-laccase while exposed at 50°C. The half-lives are 3.5, 0.8 and 1.1 h respectively while at 40°C are 8.2, 5.6 and 6.8 h respectively. **E.** Optimal temperature determination for the blue (squares), yellow (triangles) and ABTS adduct (rhombuses). **F.** Typical Arrhenius plot in the 30–60°C range for the activation energy determination.

normalized to 280 nm) shows that the typical bands of the adduct spectrum remain present. All these prove that the bond is clearly covalent, but nevertheless moderately labile and thus reversible by redox-cycling or slow hydrolysis when denatured.

Interestingly, when the yellow laccase is treated with excess dithionite, a decrease in the absorbance between 380–750 nm is noted, which is restored to only ~ 60% after reoxidation by air and desalting (Fig 4), thus behaving similar to the ABTS-adduct. This may suggest that the yellow laccase is a natural adduct formed between natural occurring small phenolic compounds and the blue form which can be obtained from the yellow form. One may expect that a heterogeneous population of such adducts exist *in vivo*, so that no definite UV-vis spectrum can be obtained, and that at least in some members of this population the blue band may additionally be quenched (reduced). The extinction coefficient of the natural adducts are also expected to be lower compared to the artificial ones. A UV-vis difference spectrum between the isolated oxidized and the dithionite reduced versions of the yellow form (Fig 4) reveals both a ~ 600 nm blue form-specific spectral feature and a ~ 460 nm guaiacol (phenolic) specific spectral feature. The 40% decrease of the phenolic spectrum after a reduction/oxidation cycle

(a–c) renders the 600 nm band more pronounced in the re-oxidized minus reduced trace (c–b). Thus, the yellow form contains a blue type 1 center, at least for a fraction of the population and/or is partially reduced.

## Enzymatic activity

Since $Q_0H_2$ is a substrate which forms the adduct with the blue form at much slower rate than the other radical forming substrates, it was employed in order to assess the catalytic and kinetic parameters of the enzyme's blue vs. yellow forms (Table 1). Remarkably, the blue-to-yellow modifications result in a distinctly higher substrate affinity (lower $K_M$), and the curves present a biphasic behavior for the adducts but not for the blue form (S6 Fig). The $K_M$ value for the higher affinity phase (physiologically relevant in magnitude) for $Q_0H_2$ is identical for the as-isolated yellow laccase form, the purple ABTS, the orange TMB, and the red guaiacol adduct forms. The apparent $k_{cat}$ values do vary somewhat if the enzyme concentrations are calculated from the absorbance at 280 nm, although these are probably altered by the adduct contribution to the UV spectrum. This proves that the adducts are not bound directly at the active site, but rather bind such that a change of the binding pocket is induced, in line with the MS findings. The adduct does not seem to function primarily as a bound electron mediator between the substrate and the type 1 Cu, since this would likely result in adduct-specific (and possibly rather high) $K_M$ values. The quenching/alteration of the type 1 spectrum is likely also due to structural alteration effect of the adduct, which may result in strain on (or fully breaking) the Cu-Cys bond, normally responsible for the blue S(p) to Cu(d) charge transfer band, such that this charge transfer is altered or rendered impossible as proved by site-directed mutagenesis of other laccases[24]. The ability to prepare both the blue form and multiple adduct forms of the protein facilitates spectroscopic and theoretical studies into the alterations, which may shed new light on the fickle nature of type 1 blue copper sites.

The blue, yellow and ABTS-laccase forms are found to show different emission fluorescence spectra when the samples are excited at 330 nm (Fig 4). The emission spectra are centered on 415 nm, 435 nm and 440 nm respectively, and are well known to be due to the type 3 copper binuclear center[33,37]. Noting that the ABTS-adduct emission spectrum is somewhat similar to that of the yellow form, the difference between these forms might be due to the conformational change caused by the covalent attachment of the product which is gating the water channel from the trinuclear site (*vide infra*).

## Enzyme stability

The stability of these three laccase adducts at high temperature was assayed and revealed that the yellow form is most stable (3.5 h at 50°C and 8.2 h at 40°C) compared to ABTS-laccase (1.1

**Table 1. Michaels-Menten parameters for three substrates for five forms of the purified laccase obtained by Eadie–Hofstee linearization in case of biphasic cases ($Q_0H_2$) and non-linear fitting using GraphPad for normal curves.** Relative standard deviations are 5–20%.

| Protein form | $Q_0H_2$ | | | | ABTS | | TMB | |
| --- | --- | --- | --- | --- | --- | --- | --- | --- |
| | Low affinity | | High affinity | | | | | |
| | $K_M$ (µM) | $k_{cat}$ (s$^{-1}$) | $K_M$ (µM) | $k_{cat}$ (s$^{-1}$) | $K_M$ (µM) | $k_{cat}$ (s$^{-1}$) | $K_M$ (µM) | $k_{cat}$ (s$^{-1}$) |
| blue laccase | 23.9 | 28.6 | - | - | - | - | - | - |
| yellow laccase | 75.5 | 7.20 | 5.40 | 3.60 | 14.1 | 5.70 | 2.70 | 3.1 |
| ABTS-laccase | 22.4 | 15.4 | 5.20 | 12.0 | 8.80 | 11.5 | 1.70 | 11.3 |
| guaiacol-laccase | 30.0 | 12.9 | 5.4 | 10.7 | 8.7 | 10.0 | 2.8 | 10.9 |
| TMB-laccase | 454 | 24.3 | 4.5 | 11.2 | 9.1 | 10.7 | 2.2 | 11.8 |

h at 50°C and 6.8 h at 40°C) and blue laccase (0.8 h at 50°C and 5.6 h at 40°C) (Fig 4). Additionally, when tested at 40°C, an activation during the first 4 h was observed as we previously shown[30]. The blue form was activated only by 11% while the ABTS adduct was activated by 16% and the yellow form by 32%. This was also found in other studies, where it was shown that the yellow laccase is more stable when exposed to ethanol, compared to the blue form[38]. The order of stability towards denaturation by guanidine (inflections points (in mM) of the GuHCl curves: yellow laccase 2360 ± 67, ABTS-laccase 2616 ± 36, blue laccase 2835 ± 57) was found to be reversed compared to that of the temperature stability: yellow laccase < ABTS-laccase < blue laccase; the temperature assay follows the activity, whereas the guanidine assay follows spectral features. On the other hand, the optimum temperature was found to be 60–65°C for all three forms, and the activation energies (by using Arrhenius plots in the 30–60°C range) were found to be very similar to each other—22.23; 21.35 and 25.29 kJ/mol for the yellow, blue and ABTS-laccase, respectively (Fig 4).

## Possible structural implications of the modified tyrosine

As previously mentioned, our laccase was successfully assigned to be the putative uncharacterized A7EM18 form *Uniprot* database[30]. Using the primary structure from this database, a 3D model was constructed by the use of $(PS)^2$-v2: Protein Structure Prediction Server[39] available at http://140.113.239.111/~ps2v2/index.php using the *Melanocarpus albomyces* laccase 3D structure (pdb code 2Q9O, one of the better-known ascomycete laccases) as template by automatic selection, sharing a 29% identity. The quality of the model was supported by values of LGscore and MaxSub (5.356 and 0.504 respectively) which classify the model as very good and good respectively and the expectation value of 2.5E-16 sustains its high reliability [40]. Much poor scores were obtained for any other templates tested. The overall structural architecture of the 3D model and the ligands involved in the active sites and their most probable coordination geometry (Fig 5, S7 and S8 Figs) suggests that the *S. sclerotiorum* laccase is a typical multi-copper oxidase. Tyr65 (uniprot numbering), identified from MS data, is located at the bottom of a surface cleft, and at the entrance of a channel to the T3 dicopper site: This channel ends at the imidazolate of the T3-ligand His133, and is flanked by the imidazoles of T3-ligands His135 and His177 (Fig 5). One may notice the peculiar sequence where the key Y65 is located: ANNTNP*Y*TNP**PNT**GV, both the repeating unit TNP or PNT and the high frequency of prolines in vicinity with asparagines and threonines. Such sequences belong to special flexible loops that are usually encountered in signaling and protein-protein interactions[41,42]. Therefore, it seems to be a connection between oxygen splitting and reactivity of the site (e.g. to promote the radical reaction). Other asco-laccases are known to be modulated by the position of the rather C-terminal tail that is gating the channel from the T2/T3 site and playing a critical role[43,44]. The distance between the Y65 and the His133 (closest ligand from the trinuclear site) is close to 13 Å, which is within the Dutton's distance window for efficient electron transfer, indication that electron modulation between the two centers could not be excluded (Fig 5). Besides, since the tyrosine is placed in between three important loops, in the center of the three cupredoxin-like motifs, it is expected that its modification induces structural changes in the whole protein—which may influence the redox and geometries of the copper ions and thus their spectral features (S7 Fig). It is interesting to mention that other Tyr residues from another laccase showed significant extra density at the ortho positions in the crystal structure and this was interpreted as evidence for nitration[45].

We have provided direct evidence for an example where a blue laccase can be converted to a yellow form *in vitro* by covalent modification at the T1 site, with metabolites produced by the laccase itself, in line with previously formulated hypotheses[10–14]. Moreover, this

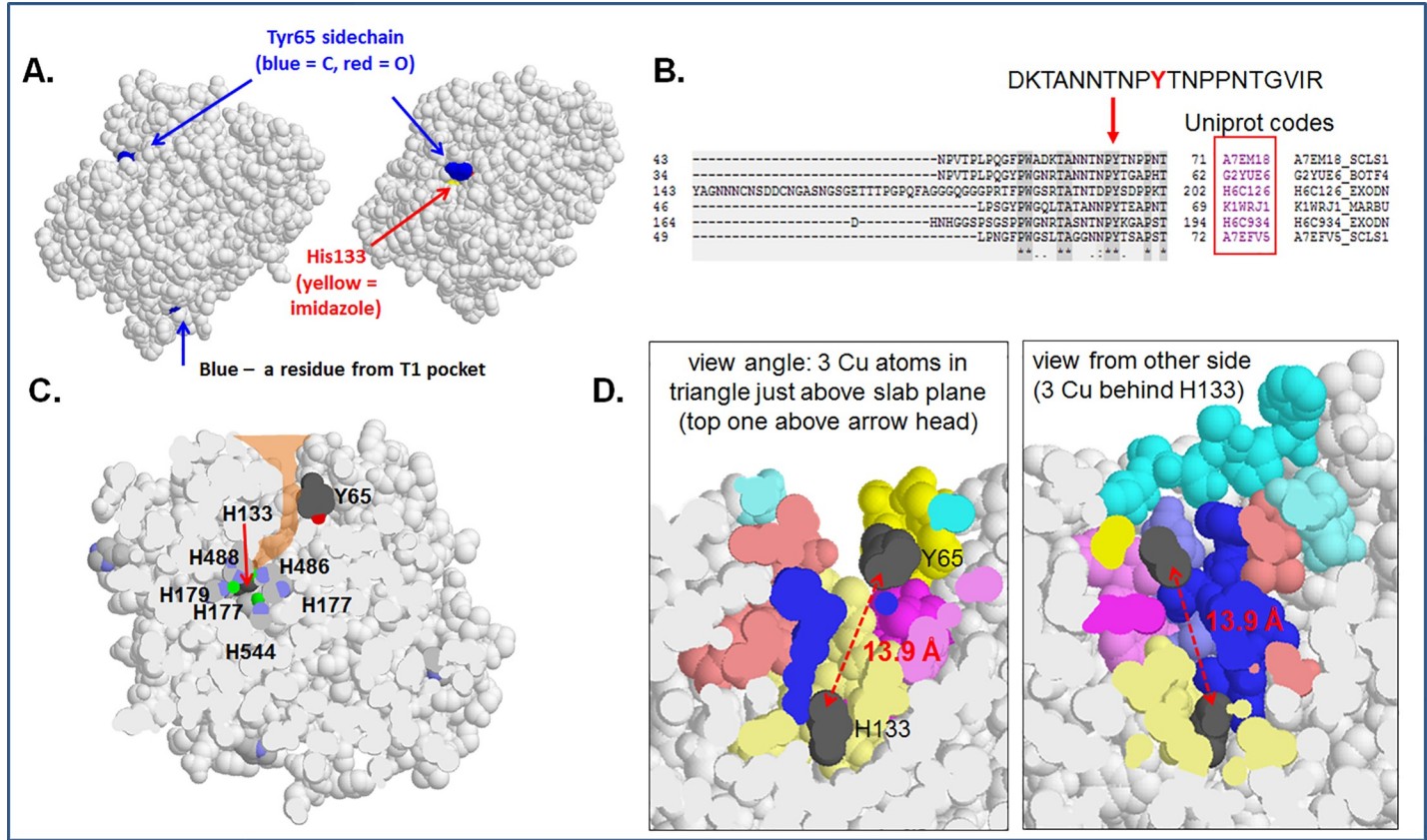

**Fig 5. Structural implications of the Y65. A.** Localization of the Tyr65 relative to the T2/3 and T1 sites. **B.** BLAST results of the key tyrosine hydrophobic loop present only in Ascomycota, showing its presence in other multicopper oxidases. The Y seems to be well conserved. **C.** Putative water channel from the trinuclear site to the protein surface and the gating position of the Y65. **D.** Distance between H133 and Y65.

autocatalytic modification significantly improves the structural and catalytic properties of the enzyme. In essence, *S. sclerotiorum* yellow laccase is an intrinsically blue multi-copper oxidase that has activated itself upon first encounter with a polyphenolic substrate.

# Supporting information

**S1 Fig. UV-vis spectra of laccase after turnover with substrates with no radical forming ability.** UV-vis spectra of blue laccase (black), blue laccase after turnover with excess of dimethanolferrocene (red) and 2,3-dimethoxy-6-methyl-1,4-benzoquinone (blue) The substrates with no radical forming ability or with extremely short life time do not form adducts in in the same reaction time scaling.
(DOCX)

**S2 Fig. Effect of nitration upon laccase adduct formation.** UV-vis spectra of nitrated laccase and ABTS-nitrated laccase compared to the normal ABTS-laccase adduct.
(DOCX)

**S3 Fig. Identification via MS of Tyr responsible for adduct formation.** ESI(+)-HRMS spectrum of the peptide that is missing in the tryptic digestion of ABTS-laccase adduct but is observed in the blue laccase digestion.
(DOCX)

**S4 Fig. HPLC-MS separation of a tyrosine-ABTS reaction mixture and identification of the reaction products.** The structures of the three identified compounds are shown in left top panel.
(DOCX)

**S5 Fig. UV-vis spectra of the free Tyr-ABTS adduct and byproduct.**
(DOCX)

**S6 Fig. Enzymatic kinetic investigations for the studied laccase adducts.** Michaelis-Menten curves for blue (A), yellow (B), ABTS (C), guaiacol (D), TMB (E) laccases and their Eadie-Hofstee linearization plots (as insets) using $Q_0H_2$ as substrate. $K_M$ and $k_{cat}$ values are listed in Table 1 from the main manuscript.
(DOCX)

**S7 Fig. A. Modelled 3D structures of the *S. sclerotinia* laccase.** Model of *S. sclerotiorum* laccase using the better-known ascomycete laccases as template (pdb code 2Q9O, *Melanocarpus albomyces*). Copper ions are depicted as orange spheres. Histidines involved in the trinuclear site are colored in yellow and the key tyrosine (Y23) identified by MS data is colored in purple (see main text). Notice the key position of the tyrosine between the three domains of the enzyme and just above the channel from the T2/T3 site. **B.** Same model with all Tyr residues indicated.
(DOCX)

**S8 Fig. Modeled active sites of *S. sclerotiorum* laccase.** The structure was modeled having *Melanocarpus albomyces* laccase as template (residue numbering is from the uniprot database entry A7EM18).
(DOCX)

## Author Contributions

**Conceptualization:** Augustin C. Mot.

**Data curation:** Niculina Hadade.

**Formal analysis:** Augustin C. Mot, Niculina Hadade, Grigore Damian, Radu Silaghi-Dumitrescu, Hendrik Heering.

**Funding acquisition:** Augustin C. Mot.

**Investigation:** Cristina Coman, Grigore Damian.

**Methodology:** Augustin C. Mot, Cristina Coman, Niculina Hadade, Radu Silaghi-Dumitrescu, Hendrik Heering.

**Project administration:** Augustin C. Mot, Hendrik Heering.

**Software:** Hendrik Heering.

**Supervision:** Augustin C. Mot, Hendrik Heering.

**Validation:** Niculina Hadade, Hendrik Heering.

**Writing – original draft:** Augustin C. Mot, Hendrik Heering.

**Writing – review & editing:** Augustin C. Mot, Niculina Hadade, Radu Silaghi-Dumitrescu, Hendrik Heering.

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
