## [Decision Letter · Decision Letter 0]

8 Oct 2019

PONE-D-19-25595

“Yellow” laccase from Sclerotinia sclerotiorum is a blue laccase that enhances its substrate affinity by forming a reversible tyrosyl-product adduct

PLOS ONE

Dear Dr Mot,

Thank you for submitting your manuscript to PLOS ONE. After careful consideration, we feel that it has interest and merit but does not fully meet PLOS ONE’s publication criteria as it currently stands. Therefore, we invite you to submit a revised version of the manuscript that addresses the points raised during the review process.

We would appreciate receiving your revised manuscript by Nov 22 2019 11:59PM. To enhance the reproducibility of your results, we recommend that if applicable you deposit your laboratory protocols in protocols.io, where a protocol can be assigned its own identifier (DOI) such that it can be cited independently in the future. For instructions see: http://journals.plos.org/plosone/s/submission-guidelines#loc-laboratory-protocols

We look forward to receiving your revised manuscript.

Kind regards,

Ligia O Martins, PhD

Academic Editor

PLOS ONE

**Journal Requirements:**

**Comments to the Author**

1. Is the manuscript technically sound, and do the data support the conclusions?

Reviewer #1: Partly

Reviewer #2: Yes

2. Has the statistical analysis been performed appropriately and rigorously? 

Reviewer #1: Yes

Reviewer #2: Yes

3. Have the authors made all data underlying the findings in their manuscript fully available?

Reviewer #1: Yes

Reviewer #2: Yes

4. Is the manuscript presented in an intelligible fashion and written in standard English?

Reviewer #1: Yes

Reviewer #2: Yes

5. Review Comments to the Author

Reviewer #1: This is a very interesting manuscript that provides novel insight as to the origin of the 'non-blue' phenotype of a class of laccases generically known as yellow or white. The authors provide solid evidence that shift in the typically CT transition typically observed for such enzyme at ~610 nm is due to tyrosyl modification by a substrate radical. I think the evidence for this conclusion is solid and along with the substance of the manuscript makes an excellent contribution to the literature on multicopper oxidases. However, the manuscript reads like a travel journal and the long explications of the ins and outs of the experimental journey shrouds the conclusions in uncertainty (what actually is the fact here?) but takes away from the actual focus and conclusions reached. I appreciate combining Results and Discussion, but if the authors want to retain this format, they HAVE TO prune the latter by AT LEAST 50% for this manuscript to be acceptable for publication.

Reviewer #2: This is a manuscript about a ‘yellow’ laccase studied aimed at showing it has been converted from normal ‘blue’ form. To attain the object ABTS has been utilized as model substrate, although it seems to be still uncertain whether the formation of adduct is the direct cause for blue laccase to be yellow. However, the present results will be publishable in PLOS ONE after revisions.

2.1.

This section is too long to write in only one paragraph.

The ratio of the absorptions at 280 nm and 600 nm is very important to show purity of protein molecules, copper content which is not shown, and redox state of copper atoms (20-24 for many MCSs). If the absorptions for UV and Vis regions have been measured with the same pathlength (Fig. 1A), it becomes 13000, too large as coming from oxidized T1Cu because the value is usually ~5,000. Optical path-length for absorption measurements has not been indicated, although concentration of protein molecule is shown. If you utilized a spectrometer with photodiode array detector, absorption in UV regions is less reliable. This is the same for the absorption ratio at 280 and 250 nm. Stronger absorption trough at 250 nm and at shorter wavelengths may suggest a deformation of protein molecule.

It is preferable to show the yields of blue and yellow laccases before and after native PAGAE if possible.

I wonder why did you did not perform hydrophobic chromatography to obtain pure blue and yellow laccases. In this study, hydrophobic chromatograph might be a preferential choice.

Possible presence of isozymes are not taken into consideration throughout study. Isozymes might readily receive modifications to turn into the yellow form.

EPR of yellow laccase shows the presence of T3Cu with an analogous content with T2Cu, suggesting a certain damage on the yellow laccase. Otherwise it TNC might be in an disordered form. EPR should be done quantification. It seems difficult to explain why the modification at the remote site gives the mixed valence Cu state.

In Fig. 1D the 330 nm band is missing and accordingly this should be discussed.

2.2

Authors write covalent attachment of ABTS via radical mechanism and the adduct is very stable but reversible. Does not this mean that the adduct form was not reversed but simply hydrolyzed?

How many Tyr residue are present on the protein surface and inside protein molecule? It would be better to show locations of every Tyr residue in the model in FigS7 and eplain why a specific Tyr residue have received the attack from substrate radicals in spite of its location at the remote site from the active site.

2.3

p. 7 L30-33 �max for ABTS-laccase and ABTS-Tyr should be shown.

p. 8 2nd paragraph Figs 2 and B simply show that ABTS and Tyr may react in a 1:1 ratio. However, it may not the same with laccase protein.

p. 8 3rd paragraph The experiment has been performed from very low pH to high alkaline pH’s, where proteins will not take the native conformation.

2.4 and 2.5

Authors seems to think the formation and decay of the adduct is passed through the same process, i. e. reversible, through the radical mechanism. However, the adduct may receive hydrolysis, giving an o-diphenol moiety or o- quinone. Even so, the modified proteins will return to blue form when fully oxidized .

2.6

Deviations should be shown in the data in Table . Even though modifications took place at remote site far from the active site, activities of enzyme derivatives are considerably affected usually, and so I never think the modification at the water channel functions to hop electrons to the trinuclear center. In addition, I don’t understand why the process is allosteric. What is the reason why the intermolecular ET process is excluded?

2.7

It is quite natural that enzymatic processes proceed faster with increasing temperature, ca. twice with increasing 10 deg, but such change in activity depending on temperature is not usuall called activation.

2.8

Degree of homology should be shown as % between your laccase and template.

Water channel is not known to transport proton, although it allows to pass through considerably bulky anions such as acetate ions and azide ion. Mutation and crystallographic studies at the wired waters involving a specific Glu in CotA and CueO have unequivocally indicated that this channel is the proton transfer pathway and the channel constructed between domains has not ability to transport protons. Differing from HCH connecting T1Cu and TNC and an His ligand coordinated to T1Cu hydrogen bonded with an acidic amino acid or covalently bonded with Trp in bilirubin oxidase, no ET pathway to reach from exterior of protein molecule to TNC is known. Accordingly, the intramolecular ET through the water channel will never take place, although an intermolecular process is not excluded.

6. PLOS authors have the option to publish the peer review history of their article (what does this mean?). If published, this will include your full peer review and any attached files.

Reviewer #1: No

Reviewer #2: No

---

## [Author Response · Author response to Decision Letter 0]

24 Oct 2019

Reviewer #1: This is a very interesting manuscript that provides novel insight as to the origin of the 'non-blue' phenotype of a class of laccases generically known as yellow or white. The authors provide solid evidence that shift in the typically CT transition typically observed for such enzyme at ~610 nm is due to tyrosyl modification by a substrate radical. I think the evidence for this conclusion is solid and along with the substance of the manuscript makes an excellent contribution to the literature on multicopper oxidases.

Reply: We thank reviewers for the encouraging comments.

 However, the manuscript reads like a travel journal and the long explications of the ins and outs of the experimental journey shrouds the conclusions in uncertainty (what actually is the fact here?) but takes away from the actual focus and conclusions reached. 

Reply: We removed some of the too phenomenological comments and readjusted the contet.

I appreciate combining Results and Discussion, but if the authors want to retain this format, they HAVE TO prune the latter by AT LEAST 50% for this manuscript to be acceptable for publication.

Reply: We remove some paragraphs and too descriptive information. It is indeed quite lengthy; however, the journal has no restrictions on the length of the manuscript, thus we did not invest too much in being very concise. However, we consider that the revised version is improved now, concerning this aspect.

Reviewer #2: This is a manuscript about a ‘yellow’ laccase studied aimed at showing it has been converted from normal ‘blue’ form. To attain the object ABTS has been utilized as model substrate, although it seems to be still uncertain whether the formation of adduct is the direct cause for blue laccase to be yellow. However, the present results will be publishable in PLOS ONE after revisions.

2.1.

This section is too long to write in only one paragraph.

Reply: We have shortened and restructured the section into multiple paragraphs.

The ratio of the absorptions at 280 nm and 600 nm is very important to show purity of protein molecules, copper content which is not shown, and redox state of copper atoms (20-24 for many MCSs). 

Reply: That is a very welcome question. Of course, the ratio of Abs at 280 nm to the wavelength of a given cofactor (type 1 copper, heme, flavin, etc.), is an indication of the purity for that protein, providing no extra altering, distortions, unknown bound cofactors. However, in the case of yellow laccase, and even converted blue form, you can’t use this without any doubts since a fully of partly bound mediator will influence the absorbance and could even keep the copper in the partly reduce form (especially, if high redox potential). For instance, the yellow form has a ratio of 125 and the blue form decrease to 34, closer but still different than others MCSs, which, as indicated are close to 22. Therefore, we included now this information in the revised manuscript, but it has to be taken with caution, we indicated that. We added in the revised manuscript the following “In addition, the ratio of the absorbances at 280 nm to 600 nm decreased from 125 (yellow form) to 34 (blue form), indicating a dramatic modification of the type 1 copper center, for the as purified laccases. This ratio is still higher than most of the multicopper oxidases (cca. 22) and has to be taken with caution, since both, the oxidation state and integrity of the type 1 copper center of the as purified forms could be fully (in yellow case) or partly (blue form) affected due to a putative bound mediator.”

If the absorptions for UV and Vis regions have been measured with the same pathlength (Fig. 1A), it becomes 13000, too large as coming from oxidized T1Cu because the value is usually ~5,000. Optical path-length for absorption measurements has not been indicated, although concentration of protein molecule is shown. If you utilized a spectrometer with photodiode array detector, absorption in UV regions is less reliable. This is the same for the absorption ratio at 280 and 250 nm. Stronger absorption trough at 250 nm and at shorter wavelengths may suggest a deformation of protein molecule.

Reply: We are very thankful to the reviewer since his question make us solve a misunderstanding in this Figure. The inset part of the Figure was from 2x and 4x more concentrated yellow and blue form, respectively compared to the full spectral range spectra. We thought that the spectral features in visible would be more obvious if more concentrated samples would be used. However, we corrected now with the exact zoomed part of the spectra so that no confusion may arise. The absorptivity of T1Cu is for our as isolated blue laccase is cca. 2700 M-1cm-1, which is a bit low compared to other similar enzymes but part structural alteration and/or incomplete oxidation form might be the cause. We used high resolution double-beams spectrophotometer with photo-multiplicator detector.

It is preferable to show the yields of blue and yellow laccases before and after native PAGAE if possible.I wonder why did you did not perform hydrophobic chromatography to obtain pure blue and yellow laccases. In this study, hydrophobic chromatograph might be a preferential choice.Possible presence of isozymes are not taken into consideration throughout study. Isozymes might readily receive modifications to turn into the yellow form.

Reply: Unfortunately, in any of our numerous trials, we did not monitor the exact mass of the fungus and the total amount of protein that we obtained. We could roughly say that we obtained about 0.1-0.5 mg of blue laccase and about 1-2 mg of yellow laccase from 10 L cultures. We did use hydrophobic chromatography, as indicated in this and our previous manuscript but that did not suffice to completely remove the polyphenols and isolate the blue form. We did think of the possibility of the isozymes but we have no evidence to support this since both different zymograms and MS analysis reveals the same unique major laccase. Possible other lower mass ones may exist but that were removed during our purification steps.

EPR of yellow laccase shows the presence of T3Cu with an analogous content with T2Cu, suggesting a certain damage on the yellow laccase. Otherwise it TNC might be in an disordered form. EPR should be done quantification. It seems difficult to explain why the modification at the remote site gives the mixed valence Cu state.

Reply: We agree that both T1 and TNC might be structurally distorted but the low protein amount of protein prevented us for more EPR exploration. However, the only fact that the yellow and blue forms have similar EPR spectra we considered to be enough for our present paper.

In Fig. 1D the 330 nm band is missing and accordingly this should be discussed.

Reply: It is not noticeable due to strong ABTS overlapping; however, we are not sure that it is completely missing.

2.2

Authors write covalent attachment of ABTS via radical mechanism and the adduct is very stable but reversible. Does not this mean that the adduct form was not reversed but simply hydrolyzed? 

Reply: It could be and the free Tyr-ABTS adduct showed a hydrolysis at higher pH. The reversibility that we mentioned does not exclude the possible hydrolysis. However, this putative hydrolysis is much slower than reversibility triggered by reduction with dithionite, or other stronger radical substrates since the adduct is stable for weeks without any visible alteration (in UV-vis spectra) but radical or reducing agent exposure leads to detectable cleavage.

How many Tyr residue are present on the protein surface and inside protein molecule? It would be better to show locations of every Tyr residue in the model in FigS7 and explain why a specific Tyr residue have received the attack from substrate radicals in spite of its location at the remote site from the active site.

Reply: We updated the FigS7 accordingly. There are 24 Tyr in the protein molecule. At this time, we can’t completely explain why this specific Tyr residue is attacked. However, the consequences are very important and probably future studies, probably including site directed mutagenesis will reveal a detailed explanation.

2.3 p. 7 L30-33 �max for ABTS-laccase and ABTS-Tyr should be shown.

Reply: done

p. 8 2nd paragraph Figs 2 and B simply show that ABTS and Tyr may react in a 1:1 ratio. However, it may not the same with laccase protein.

Reply: We agree with this. However, we performed HPLC-DAD-MS analysis on the peptides digest of the protein and find out that one specific Tyr was preferred with minor other as non-specific. In addition, it is difficult to imagine the Tyr-ABTS adduct with a different stoichiometry. The MS spectrum of the involved peptide revealed that only one specific ABTS was bound.

p. 8 3rd paragraph The experiment has been performed from very low pH to high alkaline pH’s, where proteins will not take the native conformation.

Reply: That is very true. Though the obtained pKa will have no physiological relevance, however, the protein was still soluble and allowed a minimum information regarding the spectral change for simple comparison, suggesting a similar behaviour between free Tyr adduct and protein adduct.

2.4 and 2.5

Authors seems to think the formation and decay of the adduct is passed through the same process, i. e. reversible, through the radical mechanism. However, the adduct may receive hydrolysis, giving an o-diphenol moiety or o- quinone. Even so, the modified proteins will return to blue form when fully oxidized.

Reply: It could be and the free Tyr-ABTS adduct showed a hydrolysis at higher pH. The reversibility that we mentioned does not exclude the possible hydrolysis. However, this putative hydrolysis is much slower than reversibility triggered by reduction with dithionite, or other stronger radical substrates since the adduct is stable for weeks without any visible alteration (in UV-vis spectra) but radical or reducing agent exposure leads to detectable cleavage.

2.6

Deviations should be shown in the data in Table. Even though modifications took place at remote site far from the active site, activities of enzyme derivatives are considerably affected usually, and so I never think the modification at the water channel functions to hop electrons to the trinuclear center. In addition, I don’t understand why the process is allosteric. What is the reason why the intermolecular ET process is excluded?

Reply: We agree that the allosteric effect is too speculative, together with water channel function, therefore we removed it.

2.7

It is quite natural that enzymatic processes proceed faster with increasing temperature, ca. twice with increasing 10 deg, but such change in activity depending on temperature is not usuall called activation.

Reply: We think we were a bit unclear. There were two distinct experiments that lead to results from Figure 4DE. In Figure 4E, it is indeed temperature dependence of the enzyme activity (the activity was monitored or measured at different temperatures), that’s indeed not activation. But in Figure 4D, the enzyme was incubated at different temperatures for different period of time, and from time to time an aliquote was tested for its activity at measured room temperature. Since after an incubation of the enzyme at higher temperature (measured at RT), its activity significantly increased, we considered it an activation. We rewrite it for better clarity.

2.8

Degree of homology should be shown as % between your laccase and template.

Water channel is not known to transport proton, although it allows to pass through considerably bulky anions such as acetate ions and azide ion. Mutation and crystallographic studies at the wired waters involving a specific Glu in CotA and CueO have unequivocally indicated that this channel is the proton transfer pathway and the channel constructed between domains has not ability to transport protons. Differing from HCH connecting T1Cu and TNC and an His ligand coordinated to T1Cu hydrogen bonded with an acidic amino acid or covalently bonded with Trp in bilirubin oxidase, no ET pathway to reach from exterior of protein molecule to TNC is known. Accordingly, the intramolecular ET through the water channel will never take place, although an intermolecular process is not excluded.

Reply: There is a 29% identity in the sequence. It is however the best available templete. We remove the the too speculative discussion regarding proton/ET transfer via the water channel.

---

## [Decision Letter · Decision Letter 1]

7 Nov 2019

“Yellow” laccase from Sclerotinia sclerotiorum is a blue laccase that enhances its substrate affinity by forming a reversible tyrosyl-product adduct

PONE-D-19-25595R1

Dear Dr. Mot,

We are pleased to inform you that your manuscript has been judged scientifically suitable for publication and will be formally accepted for publication once it complies with all outstanding technical requirements.

With kind regards,

Ligia O Martins, PhD

Academic Editor

PLOS ONE

Additional Editor Comments (optional):

Reviewers' comments:

Reviewer's Responses to Questions

**Comments to the Author**

1. If the authors have adequately addressed your comments raised in a previous round of review and you feel that this manuscript is now acceptable for publication, you may indicate that here to bypass the “Comments to the Author” section, enter your conflict of interest statement in the “Confidential to Editor” section, and submit your "Accept" recommendation.

Reviewer #1: All comments have been addressed

Reviewer #2: All comments have been addressed

2. Is the manuscript technically sound, and do the data support the conclusions?

Reviewer #1: Yes

Reviewer #2: Yes

3. Has the statistical analysis been performed appropriately and rigorously? 

Reviewer #1: Yes

Reviewer #2: Yes

4. Have the authors made all data underlying the findings in their manuscript fully available?

Reviewer #1: Yes

Reviewer #2: Yes

5. Is the manuscript presented in an intelligible fashion and written in standard English?

Reviewer #1: Yes

Reviewer #2: Yes

6. Review Comments to the Author

Reviewer #1: The authors have improved the manuscript in response to reviewer comments. There are no further concerns.

Reviewer #2: The manuscript has been almost properly revised according to my questions and comments, while future problems to be solved have been left.

7. PLOS authors have the option to publish the peer review history of their article (what does this mean?). If published, this will include your full peer review and any attached files.

Reviewer #1: No

Reviewer #2: No

---

## [Editor Report · Acceptance letter]

9 Jan 2020

PONE-D-19-25595R1 

“Yellow” laccase from *Sclerotinia sclerotiorum* is a blue laccase that enhances its substrate affinity by forming a reversible tyrosyl-product adduct 

Dear Dr. Mot:

I am pleased to inform you that your manuscript has been deemed suitable for publication in PLOS ONE. Congratulations! Your manuscript is now with our production department. 

With kind regards,

on behalf of

Dr Ligia O Martins 

Academic Editor

PLOS ONE